# Towards Enhanced Osteointegration: A Comparative and In-Depth Study of the Biocompatibility of an Innovative Calcium-Doped Zirconia Coating for Biomedical Implants

**DOI:** 10.3390/jfb16060191

**Published:** 2025-05-22

**Authors:** Tchinda Alex, Olivier Joubert, Richard Kouitat-Njiwa, Pierre Bravetti

**Affiliations:** Université de Lorraine, CNRS, IJL, F-54000 Nancy, France

**Keywords:** biocompatibility, Ca-SZ coating, cytokines, osteogenesis, osseointegration

## Abstract

Innovation in oral implantology is constantly on the move, with a constant search for new biomaterials to overcome many of the limitations of the biomaterials used in current implantable medical devices. This study explores the biocompatibility of an innovative 5% calcium-to-zirconia (Ca-SZ) coating deposited by PVD on TA6V substrates for use in oral implantology. In order to determine the contribution of the Ca-SZ coating, an in vitro biocompatibility study was carried out to assess the potential influence of the Ca-SZ coating (1) on the viability and proliferation of saos-2 and HaCaT cells over a short-term exposure period of 96 h, (2) on the synthesis of pro-inflammatory cytokines, and (3) on the synthesis of osteogenic differentiation markers over a long-term exposure period of 21 days, in comparison with reference biomaterials. The sampling consisted of *n* = 3 biological replicates, and a *p*-value of <0.05 was used as the threshold for statistical significance. Viability and proliferation kinetics to WST-1 and CyQUANT NF, respectively, showed improved viability/proliferation of Ca-SZ exposed to both cell lines independently. The TNF-alpha and IL-6 assays revealed reduced levels of cytokines compared with the reference biomaterials, including the control groups. In parallel, in Saos-2 cells exposed to Ca-SZ for 21 days under osteogenic conditions increased expression of osteogenic markers, such as the synthesis of soluble collagens, alkaline phosphatase (ALP), osteopontin, and osteocalcin, reflecting dynamic and facilitated osteoblastic differentiation, which was supported by the formation of hydroxyapatite (HA) crystals observed by SEM micrograph and confirmed by EDS mapping. In conclusion, Ca-SZ demonstrates an overall better biocompatibility compared with reference biomaterials, linked to a bioactive interaction of calcium, promoting cell proliferation and differentiation for optimal osteointegration, underlining its potential as a relevant innovation for next-generation implants.

## 1. Introduction

The use of medical implants made from titanium alloys such as TA6V in particular is a widely-accepted approach in implant surgery in general and, particularly, in oral implantology due to its toughness, which is favorable to biomedical applications involving mechanical stresses, and its recognized biocompatibility, which has benefited from significant clinical experience [1]. However, despite their acknowledged clinical success, titanium alloy dental implants are increasingly facing major long-term clinical challenges. For example, although titanium dental implants are generally well tolerated by bone tissue, the latter is susceptible to corrosion due to degradation of the oxide layer, resulting in the release of metal particles and promoting chronic inflammation and peri-implant osteolysis [2,3]. This peri-implant tissue necrosis can lead to loosening of the dental implant [4]. Peri-implantitis caused by bacteria is currently one of the major causes of these failures, especially as studies suggest a prevalence rate of between 12% and 43% of implants after 5 to 10 years of implantation, an observation that is constantly increasing [5]. These multi-source predispositions highlight the need and interest in developing new advanced biomaterials to improve biocompatibility and optimize osseointegration.

Among the candidate materials being considered for improving implants, zirconia in particular is attracting growing interest due to its bio-inertia, proven biocompatibility, and ideal aesthetic appearance [6]. However, it should be emphasized that zirconia has disadvantages such as low toughness due to its ceramic nature, unlike TA6V. Consequently, with the aim of overcoming these drawbacks, recent research is focusing on hybrid biomaterials that combine the biological properties of zirconia with bioactive elements capable of inducing directed cellular responses [7]. Consequently, the addition of calcium to zirconia (Ca-SZ) is a promising approach. Calcium, as a natural component of the bone matrix, plays an essential role in the differentiation and proliferation of bone cells, also acting as a chemotactic agent promoting the attraction and adhesion of osteoblasts responsible for bone remodeling [8,9].

Basically, osseointegration is the colonization and anchoring of an implant surface by the surrounding bone tissue. This new bone formation is based on complex cellular processes, mediated by molecular interactions between bone cells and molecules present on the implant surface. Numerous studies suggest that calcium stimulates and promotes cellular responses dedicated to the osseointegration process, which is essential for lasting healing [10]. In addition, some authors report that calcium ions stimulate the secretion of proteins such as osteopontin and osteocalcin, which are major markers of bone maturation and mineralization. These proteins contribute to the formation of the peri-implant bone matrix, guaranteeing long-term osseointegration [11,12,13]. More interestingly, the recent literature suggests that calcium plays an immunomodulatory role by attenuating cellular responses dedicated to the synthesis of pro-inflammatory cytokines at the bone–implant interface. This has the essential benefit of limiting the consequences underlying an acute and chronic inflammatory response, thereby contributing to optimal osseointegration [14,15,16]. Furthermore, the presence of zirconia in the Ca-SZ coating plays an important role in providing an effective shield to limit bacterial biofilm colonization and proliferation, thereby minimizing the risk of peri-implantitis [17]. The vulnerability of dental implants arises from the absence of root cementum and periodontal ligament as protective systems between the transgingival tissues and the dental implant, unlike the natural tooth, which constitutes the main point of bacterial infiltration. Once again, this approach is part of a set of common, diverse, and varied actions necessary for the management of peri-implant disease, ranging from the bactericidal/bacteriostatic effects induced by the intrinsic properties of certain zirconium biomaterials to the release of certain ions/agents by the surface functionalization, to the external driving forces [18].

In this context, this study aims to evaluate the biocompatibility and cellular response of a 5% Ca-SZ coating produced by physical vapor deposition (PVD) on TA6V discs to take advantage of the biocompatibility of zirconia combined with calcium on the surface and the ductility of the titanium substrate for future application in dental implantology.

This study incorporates a range of techniques to assess the in vitro biocompatibility of the Ca-SZ coating by measuring the viability and proliferation of bone (Saos-2) and epithelial (HaCaT) cells using kinetics over a 4-day exposure period. This is followed by an analysis of cell adhesion morphology using SEM and the integrity of the cytoskeleton of Saos-2 cells using confocal microscopy, two key indicators of biomaterial compatibility. Secondly, an assessment of the early and late influence of the coating on the modulation of the synthesis of the main inflammatory markers TNF-α and IL-6 is carried out. Similarly, an assessment of the early and late influence of the coating in osteogenic conditions on the cellular response through the synthesis of the main markers of osteogenesis (alkaline phosphatase, soluble collagen, osteopontin, and osteocalcin) is conducted. These analyses will provide a profile of the reliability of this innovative biomaterial and its impact on the bone remodeling process.

## 2. Materials and Methods

### 2.1. Sample Preparation and Sterilisation

As a reference, ZrO_2_/Y_2_O_3_-98/2at% (simplified name: YSZ) and ZrO/CaO-95/5w% (simplified name: CSZ) discs, Lot No. 103663, were used to assess the contribution of the Ca-SZ coating compared with the solid material. All the D20 × 2 mm discs were obtained from Ampere Industrie^®^ (Saint-Ouen-l’Aumône, France). TA6V discs of the same dimensions were obtained from VISY Implant^®^ (Chavanod, France), on which the Ca-SZ coating was produced by physical vapour deposition (PVD) (simplified name: Ca-SZ) and then characterized by the authors [19]. Bare TA6V discs were also used to compare the contribution of the Ca-SZ coating. Each disc was first cleaned by immersion for 10 min in 70% ethanol obtained from VWR International^®^ (Rosny-sous-bois-Cedex, France), then in distilled water, and last in osmosed water. The whole assembly was then sterilized by autoclaving using the ‘prions’ program at 134 °C and 2.1 Bar before use in cell culture.

### 2.2. Cell Cultures

Two cell lines were used: Saos-2 cells (bone line) and HaCaT cells (human epithelial line). Saos-2 bone cells derived from osteosarcoma in an 11-year-old Caucasian girl, were obtained from the American Type Culture Collection (ATCC^®^ Manassas, VA, USA). This is an adherent cell line. These cells are cultured in McCoy (modified) 5A medium (Gibco™, Fisher Scientific, Illkirch, France) supplemented with 1% penicillin/streptomycin, 0.05% amphotericin B, 2% of L-glutamine, and 15% foetal calf serum (Sigma-Aldrich^®^, Saint-Quentin-Fallavier, France), at 37 °C with 5% CO_2_ under 90% humid atmosphere. Cells are seeded at 1 million in T75 flasks (75 cm^2^) at a density of 100,000 cells/mL. The cells are passaged every 2 to 3 days, depending on the confluence. HaCaT cells (Cat N°: 300493) are a human cell line derived from keratinocytes of a 62-year-old spontaneously immortalized Caucasian male obtained from Cytion GmbH (Eppelheim, Germany). This adherent line was cultured in DMEM recommended culture medium (Gibco™, Fisher Scientific) supplemented with 1% penicillin/streptomycin, 0.05% amphotericin, 2% L-glutamine, and 10% fetal calf serum (Sigma-Aldrich^®^, Saint-Quentin-Fallavier, France). Cultures were incubated in a humidified atmosphere at 37 °C with 5% CO_2_ under a 90% humid atmosphere. Cells were seeded at 800,000 in T75 flasks (75 cm^2^). The cells were passaged every 2 to 3 days, depending on the confluence. For all the tests performed, the control consists of a cell seeding in a tissue culture-treated polystyrene well (PS). All experiments were performed with cells at passages 19–27.

For long-term culture over 21 days, Saos-2 cells were seeded at a concentration of 40,000 cells/mL in 12-well plates (Sarstedt, Numbercht, Germany), with 2 mL of cell suspension per well in an arrangement of N = 3 independent biological replicates. The culture medium was renewed every three days to maintain optimal growth conditions.

Cultures for pro-inflammatory cytokine analysis were grown in modified McCoy 5A standard medium (Gibco™, Fisher Scientific). Supernatants were collected on days 3, 7 and 21. Each supernatant sample was clarified by centrifugation at 1500× *g* for 10 min to remove cellular debris. The clarified supernatants were then transferred to clean tubes, taking care not to disturb the pellets, and stored at −80 °C for subsequent analysis.

For the assessment of osteoblast differentiation, parallel cultures of Saos-2 were established at the same cell density, 40,000 cells/mL, 2 mL/well, and incubated in osteoblast differentiation medium (Osteoblast Differentiation Medium, Cell Applications, Inc. Lot: 76604, San Diego, CA, USA) supplemented with 1% penicillin/streptomycin, 0.05% amphotericin B, 2% L-glutamine, and 15% fetal calf serum (Sigma-Aldrich^®^, Saint-Quentin-Fallavier, France) in an arrangement of N = 3 independent biological replicates. Supernatants were collected on days 7, 14, and 21 for the assessment of differentiation markers. These supernatants were clarified and stored under the same conditions as described above to ensure analyte stability and assay reliability.

### 2.3. Cell Viability Kinetics

Saos-2 and HaCat cells were seeded in 12-well plates at 40,000 cells/mL at 2 mL per well in the presence of TA6V, CSZ; YSZ, and Ca-SZ discs and controls in an arrangement of N = 3 independent biological replicates for 24 h, 48 h, and 96 h. After each respective exposure time, the medium was aspirated from each well, and the wells were washed twice with 1 mL of SVF-free culture medium. 200 µL (10% of the final well volume) of tetrazolium salt solution (WST-1) (Roche Diagnostics GmbH, Mannheim, Germany) was added to each well containing 2 mL of fresh SVF-free medium, protected from light. The plates were incubated for a further 3 h at 37 °C. A technical triplicate of 170 µL of cell suspension was taken from each well and placed in a new 96-well plate (Sarstedt, Numbercht, Germany). The optical density absorbance was read at 450 nm to quantify the metabolic activity of the cells using a plate reader (FLUOstar Omega, Nancy, France).

### 2.4. Cell Proliferation Kinetics

Saos-2 cell proliferation was quantified by the CyQUANT^®^ NF assay (Invitrogen^®^, Villebon-sur-Yvette, France) on cells grown in the presence of TA6V, CSZ; YSZ, and Ca-SZ discs in a kinetic arrangement of N = 3 independent biological replicates. After incubation time (24 h, 48 h, 96 h), the cells were washed twice with PBS (Sigma-Aldrich^®^, Saint-Quentin-Fallavier, France). The cells were detached using 600 µL of trypsin EDTA (Sigma-Aldrich^®^, Saint-Quentin-Fallavier, France) in each well, then harvested and centrifuged at 130× *g* for 10 min. The pellet was then resuspended in 305 µL of CyQUANT NF binding solution. The whole batch was homogenized and incubated at 37 °C for 45 min according to the manufacturer’s recommendations. Fluorescence was measured at an excitation wavelength of 485 nm and an emission wavelength of 520 nm using a plate reader (FLUOstar Omega).

### 2.5. Morphology and Cell Adhesion by SEM

Saos-2 bone cells were seeded in the presence of TA6V, CSZ, YSZ, and Ca-SZdiscs of at 25,000 cells/mL and 4 mL per well under the same conditions as the proliferation follow-ups. The cells in the presence of the discs were then incubated at 37 °C with 5% CO_2_ under a 90% humid atmosphere for 96 h.

Then, 96 h after exposure, the medium was aspirated from each well and the wells were washed twice with 1 mL of phosphate buffer. The cells were then fixed on each of the 4 different discs by adding a suspension of 800 µL of 5% glutaraldehyde (Sigma-Aldrich^®^, Saint-Quentin-Fallavier, France) and then stored at 4 °C protected from light for 30 min. The discs were then washed 3 times in PB, and 800 µL of 1% osmium tetroxide (Sigma-Aldrich^®^, Saint-Quentin-Fallavier, France), an oxidizing agent that completes the action of the fixative, was added in the dark for 30 min. The discs were washed twice with distilled water and then dehydrated by a gradual concentration step of ethanol at 35%, 50%, 75%, twice 95%, and 2 times 100%, with a waiting time of 15 min between each step. A total of 1 mL of hexamethyldisilazane (HMDS) (Sigma-Aldrich^®^, Saint-Quentin-Fallavier, France) was added to the disks for 10 min. The HMDS was aspirated, and a further 1 mL of HMDS was deposited. The plate containing the discs was placed in a vacuum desiccator for 12 h. The disks were then metallised by depositing a 15 nm thick gold layer using the metallizer (Safematic Compact Coating unit-010, Nancy, France) under pressure for 10 min. The disks were then observed on the Quanta™ FEG 650 SEM (Institut Jean Lamour, Nancy, France) at 5.00 kV.

### 2.6. Observation of the Actin Cytoskeleton by Confocal Laser Scanning Microscopy

Cells grown on disks at 50,000 cells/mL for a volume of 3 mL per well were fixed and permeabilized for actin labelling with phalloidin. After 72 h, the cells were fixed at room temperature (22 °C) for 30 min with 800 µL Antigenfix (DiaPath^®^; Lot 2021X05187; Martinengo, Italy) after rinsing. Cells were then rinsed with 1 mL of PBS before the addition of 800 µL of 0.2% Triton X-100 (Sigma-Aldrich^®^, Lot SLCF6075, Saint-Quentin-Fallavier, France) in PBS for 15 min at room temperature. Cells were incubated for 1 h at room temperature in 1% BSA in PBS (Sigma-Aldrich^®^, Saint-Quentin-Fallavier, France), and then, 800 µL of Alexa Fluor 488 anti-actin phalloidin solution (Invitrogen^®^, Thermo Fisher Scientific (Waltham, MA, UAS), Ref: A12379), diluted 1:200 in 1% BSA in PBS, was added to each well after rinsing. The samples were incubated overnight at 4 °C. Confocal observations of the samples were performed using a Leica TCS SP5-X confocal microscope (Leica Microsystems, Wetzlar, Germany), equipped with a super-continuous NKT laser source (White Light Laser, WLL, Moscow, Russia). Labelled samples were excited at 488 nm, with fluorescence emission collected above 515 nm using a high-pass filter. Images were acquired in sequential mode at a resolution of 512 × 512 pixels (scale bar and dimension on images), under a 40× water immersion objective, AU 1 pinhole aperture. The z-pitch was optimized to around 0.5 µm. The laser power and detector gain were adjusted to maximize the signal while reducing noise, including photobleaching.

### 2.7. Determination of Pro-Inflammatory Cytokine Synthesis

The concentration of soluble TNF-alpha was measured in culture supernatants on D3, D7, and D21 using the Human ELISA TNF-alpha kit (Invitrogen^®^ Thermo Fisher Scientific; Lot: 2403-3219) from 100 µL of cell supernatant clarified according to the manufacturer’s strict recommendations. Similarly, the concentration of soluble IL-6 was determined at the same sampling times, using the Human IL-6 instant ELISA kit (Invitrogen^®^, Thermo Fisher Scientific Lot: 413742-001) here; a low control named “control N” was included in the kit by the manufacturer to compare our samples. Assays were performed according to the manufacturer’s instructions, using 50 µL of clarified cell supernatant. Absorbance was read at 450 nm using a plate reader (FLUOstar Omega).

### 2.8. Soluble Collagen Assay

The concentration of all soluble collagens was measured in the supernatants of Saos-2 cell cultures in osteogenic medium using the Sircol 2.0^TM^ kit (Biocolor^®^; Belfast BT3, UK) on D3, D7, and D21. A total of 175 µL of Sircol staining reagent was added to 100 µL of each clarified supernatant sample in 96-well microplates. The plate blank and dye release reagents were used as a control in the presence of collagen in the solution. The whole plate was incubated for 30 min under agitation at 300 rpm after sealing to promote the formation of collagen–dye complexes. The plate was then centrifuged at 350× *g* for 120 min to precipitate the stained collagen. The supernatant was carefully aspirated to preserve the collagen deposit at the bottom of the wells; then, each well was washed with 250 µL of wash reagent and centrifuged at 350× *g* for a further 60 min. After aspiration of the supernatant, 200 µL of dye-release reagent was added to each well, followed by shaking at 700 rpm for 30 min. Absorbance was read at 556 nm using a microplate reader (FLUOstar Omega). The concentration of soluble collagen was determined by comparison with standards.

### 2.9. Alkaline Phosphatase Activity Measurement

Alkaline phosphatase (ALP) activity was measured in the supernatants of Saos-2 cells cultured in osteogenic medium at time points D3, D7, D14, and D21 in the presence of biomaterials. The assay was carried out using the PAL-200 kit obtained from LIBIOS S.A.S (Pontcharra Sur Turdine, France; Lot: C082.24A) according to the manufacturer’s instructions, using 20 µL of freshly collected supernatant throughout the culture period following kinetics at 5 min and 20 min incubation. The absorbance values obtained were measured at 405 nm using a microplate reader (FLUOstar Omega).

### 2.10. Osteocalcin and Osteopontin Assays

The concentration of soluble osteocalcin was measured in the culture supernatants of Saos-2 cells cultured in osteogenic medium on D3, D7, and D21 using the human osteocalcin instant ELISA kit (Invitrogen™ Thermo Fisher Scientific; Lot: 298292-010) from 25 µL of cell supernatant clarified according to the manufacturer’s strict recommendations. Similarly, the concentration of soluble osteopontin was determined at the same sampling times using the human osteopontin ELISA kit (Invitrogen^®^ Thermo Fisher Scientific; Lot: 388090-003). Assays were performed according to the manufacturer’s instructions using 20 µL of clarified cell supernatant. Absorbance was read at 450 nm using a plate reader (FLUOstar Omega).

### 2.11. Observation of Hydroxyapatite Crystals by SEM

After 21 days of cell culture in osteogenic medium, the samples were air-dried, then coated with a thin layer of 15 nm carbon by a metallizer (Safematic Compact Coating unit-010) before being observed by SEM at 10.00 kV. EDS mapping was carried out between 5 and 10.00 kV on the Quanta™ FEG 650 SEM (Institut Jean Lamour, Nancy, France).

### 2.12. Statistical Analysis

Statistical analyses were performed using the XRealStats version 1 extension in Excel. For each exposure time, we used a one-way analysis of variance (ANOVA) to assess differences between groups. Then, multiple comparisons between pairs of groups were performed using the Tukey HSD/Kramer post hoc test, applying the Bonferroni correction to reduce the risk of obtaining falsely significant results in multiple comparisons. This correction adjusts the significance threshold by making it stricter, which limits the probability of detecting differences due to repeatability.

Differences were considered significant for a value of *p* < 0.05, with the levels of significance indicated by the following symbols: * for 0.05 ≤ *p* < 0.1; ** for 0.01 ≤ *p* < 0.05, and finally *** for 0.001 and below ≤ *p* < 0.01.

## 3. Results

### 3.1. Cellular Viability Kinetics

Figure 1 shows the viability kinetics of Saos-2 cells cultured on Ca-SZ, CSZ, TA6V, and YSZ compared to the 24 h, 48 h, and 96 h exposure using the WST-1 assay. The absorbance measured is directly proportional to the metabolic activity of the cells, indicating their viability. At 24 h, Saos-2 cells showed significant differential viability, particularly in the presence of the Ca-SZ coating, significantly outperforming the TA6V group and the YSZ group. At 48 h, an overall increase in viability was observed for all biomaterials compared with the controls. Here, the Ca-SZ group showed significantly more cellular metabolic activity than the other reference biomaterials. At 96 h, DO values remained high and relatively comparable, apart from the Ca-SZ coating, which was superior to the TA6V.

Figure 2 illustrates the viability kinetics of HaCaT cells grown on Ca-SZ, CSZ, TA6V, and YSZ compared to the 24 h, 48 h, and 96 h exposure using the WST-1 assay. The absorbance measured is directly proportional to the metabolic activity of the cells, indicating their viability. At 24 h, HaCaT cells showed significant metabolic activity in the presence of all the biomaterials. It should be noted that the DO values were significant for cells in the presence of the Ca-SZ coating, outperforming the control. At 48 h, an overall increase in viability was observed for all biomaterials compared with controls. Once again, the Ca-SZ group showed a significantly more dominant cellular metabolic activity, outperforming the TA6V, CSZ, YSZ, and control groups. At 96 h, DO values increased with the number of cells present. Although cell viability in the presence of the Ca-SZ coating was significantly higher than with TAV6, it was still relatively comparable to the other reference biomaterials.

### 3.2. Cell Proliferation Kinetics

The cell proliferation kinetics of Saos-2 cells grown on Ca-SZ, CSZ, TA6V, and YSZ, compared to the 24 h, 48 h, and 96 h exposure by CyQUANT NF, is presented in Figure 3. Here, the relative fluorescence unit (RFU) is directly proportional to the amount of DNA in the cells, indicating the evolution of the cell number. At 24 h, Saos-2 cells were relatively numerous in the presence of all the biomaterials. RFU values were significant for cells in the presence of the Ca-SZ coating, outperforming the CSZ, TA6V, and YSZ groups. The 48 h exposure time showed an overall significant relative increase in cell proliferation observed for all biomaterials compared to controls. Although cell proliferation in the presence of the Ca-SZ coating was significantly higher than in the TAV6 group, this was similar to the other reference biomaterials. At 96 h, the RFU values increased with the increasing number of cells present, with no significant differences between the biomaterials.

### 3.3. Morphology and Cell Adhesion by SEM

SEM micrographs show dense organizations of Saos-2 cells (magnification ×200, ×500, and ×1000) grown on TA6V, CSZ, and YSZ surfaces and the Ca-SZ coating after the 96 h exposure (Figure 4). The cells exhibited a spreading and adherent morphology, suggesting a good affinity with the aforementioned substrates. Membrane extensions, such as pseudopodia and filopodia, are also clearly visible, suggesting strong cell adhesion and the establishment of strong intercellular connections. The ×500 and ×100 magnifications reveal cell surfaces covered with multiple spicules and microvilli, typical of this cell line. The general organization of the cell carpet is homogeneous, with complete coverage of the surfaces of the different biomaterials and an oriented arrangement along the grooves of the subjacent surfaces of the TA6V group (Figure 4a–c) and the Ca-SZ coating (Figure 4d–f). In contrast, the cells are arranged randomly on the CSZ and YSZ surfaces.

### 3.4. Observation of Actin Cytoskeleton Fluorescence by Confocal Laser Scanning Microscopy

Confocal microscopy analysis of Saos-2 cytoskeletal actin, exposed to the surfaces of TA6V, CSZ, YSZ, and the Ca-SZ coating for 72 h, reveals a dense and cohesive organization of the actin cytoskeleton (Figure 5). The cells exhibit a spreading and adherent morphology, confirming the previous SEM observation, this time with the presence of a dense network of longitudinally oriented actin filaments parallel both to each other and to the surface of their respective substrates. The distribution of the actin network is even more evident at high magnification in TA6V, CSZ, YSZ, and the Ca-SZ coating, with no significant differences between the biomaterials, reflecting the high integrity of the actin cytoskeleton (Figure 5c,f,i,l).

### 3.5. Determination of Pro-Inflammatory Cytokine Synthesis

#### 3.5.1. IL-6 Assay

Analysis of IL-6 levels in culture supernatants from Saos-2 cells exposed to TA6V, CSZ, and YSZ biomaterials and the Ca-SZ coating. On D3, the IL-6 level remained low (<5 pg/mL) for all the biomaterials, compared with the normal ‘control N’, which showed a much higher IL-6 concentration of around 30 pg/mL (Figure 6). The TA6V group, nevertheless, showed a significant IL-6 level compared with the control. However, levels on D7 were slightly higher than on D3 for all biomaterials, with no significant difference between them. Conversely, on D21, a slight increase in IL-6 synthesis was observed in cells exposed to the control, TA6V, CSZ, and YSZ groups, although this did not reach the level of the normal control. Importantly, the IL-6 content in cells exposed to the Ca-SZ coating not only remained stable over time between D7 and D21 but was significantly lower than the IL-6 content in cells exposed to all the biomaterials tested, although it was already lower than the normal control. This observation suggests a moderate secretion profile of cells in contact with the Ca-SZ coating, reflecting a minimal early and late inflammatory response to this biomaterial.

#### 3.5.2. TNF-α Assay

Analysis of TNF-α concentrations in the supernatants of cultures of Saos-2 cells exposed to different biomaterials (TA6V, Ca-SZ, CSZ, and YSZ) shows a significant differential synthesis of TNF-α in all biomaterials on D3 compared with the control group (Figure 7). This observation was particularly pronounced in the CSZ group. Although higher than the control, it should be noted that the level of TNF-α for cells exposed to Ca-SZ and the YSZ group was similar and remained significantly lower compared to other biomaterials. In contrast, the level of TNF-α decreased on day 7 with no significant difference between different biomaterials. This level remained relatively stable until day 21 for all the biomaterials tested, with a significant difference between all the biomaterials. Overall, the concentration of TNF-α on day 3 is the highest for the CSZ group, followed by the TA6V group, while Ca-SZ shows moderate levels at each exposure time, reflecting the most minimal inflammatory response among the biomaterials tested.

### 3.6. Osteogenic Differentiation

#### 3.6.1. Soluble Collagens Assay

The graph in Figure 8 shows the concentrations of soluble collagens (μg/mL) measured in the cell supernatant of Saos-2 cells exposed to different biomaterials (TA6V, Ca-SZ, CSZ, and YSZ) on days 7, 14, and 21. On day 7, the control and Ca-SZ groups showed significantly higher levels of soluble collagen, followed by the YSZ group. The TA6V and CSZ groups had significantly lower levels compared with the control and Ca-SZ groups. On D14, an increase in concentrations was observed for TA6V and a slight increase for CSZ, whereas the control and YSZ groups, as well as Ca-SZ, showed a decrease compared to D7 while still showing significantly lower values than the Ca-SZ group, which showed a peak and significantly dominant increase around 203 µg/mL compared to all other biomaterials. On D21, the Ca-SZ groups showed a slight decrease compared with D14 while remaining significantly higher than the TA6V and at a level similar to the control, CSZ and YSZ groups, which, nevertheless, showed an increase compared with D14. This observation suggests that Ca-SZ offers particularly favorable early pro-osteogenic conditions while remaining relatively stable over time compared with the reference biomaterials.

#### 3.6.2. Alkaline Phosphatase Activity Measurement

The graph in Figure 9 shows the alkaline phosphatase (ALP) activity measured from the cell supernatant of Saos-2 cells exposed to various biomaterials (TA6V, Ca-SZ, CSZ, and YSZ) on D3, D7, D14, and D21. PAL activity was both immediate and maximal on D3, while remaining similar for all biomaterials, with values above 100 U/L with no significant difference. On D7, a decrease in activity was observed for all groups, with values varying between 20 for the control group and 40 U/L for the YSZ group. The Ca-SZ group showed significantly higher activity than the TA6V and control groups and lower activity than the CSZ and YSZ groups. On D14, PAL activity continued to decrease, reaching values below 20 U/L for all biomaterials, illustrating a marked decrease for Ca-SZ compared with the TA6V group. On D21, a slight increase in activity was observed for all groups, with values varying between 10 and 30 U/L, like the control group, except for the TAV6 group, which was significantly lower than the control group. The decrease in PAL activity after D3 in all groups reflects an early high initial response followed by the stabilization of enzyme activity.

#### 3.6.3. Osteocalcin Assay

The graph in Figure 10 shows the osteocalcin concentration measured in the cell supernatant of Saos-2 cells exposed to TA6V, CSZ, YSZ, and Ca-SZ coating on days 7, 14 and 21. On day 7, osteocalcin concentrations were similar between the control and Ca-SZ groups, but the latter were significantly higher than the TA6V and YSZ groups. On the other hand, there was a general decrease in osteocalcin concentration for all groups on D14, particularly for the Ca-SZ group and the control group, both of which were significantly lower than the CSZ, TA6V, and YSZ groups. On D21, osteocalcin concentrations were lowest for all groups, with values close to 10 ng/mL for all biomaterials tested, although the Ca-SZ and TA6V groups were significantly higher than the control group. These results suggest that the biomaterials tested influence osteocalcin synthesis by Saos-2 cells differentially and favorably, and particularly markedly for the Ca-SZ group, which induces a significant early cellular response.

#### 3.6.4. Osteopontin Assay

Measurements of osteopontin concentration show a significant dependence on biomaterials and exposure time (Figure 11). On D7, all groups showed significantly elevated osteopontin concentration levels compared to the control. More importantly, the Ca-SZ group showed a significantly dominant osteopontin concentration level compared to the YSZ group. On D14, a moderate increase in osteopontin concentration was observed in all biomaterials. There was a significant increase on D21 of over 60 ng/mL, which was similar for all the biomaterials tested. The data clearly show a progressive increase in osteopontin concentrations over time, suggesting a dynamic cellular response, particularly for Ca-SZ, which shows the highest concentration level on D7, suggesting that this material promotes an early and intense cellular response to osteopontin synthesis compared with the TA6V, CSV, and YSZ groups.

### 3.7. SEM Observation of Hydroxyapatite Crystals

Scanning electron micrographs (SEM) reveal a complex structure of hydroxyapatite (HA) crystals formed after differentiation of Saos-2 cells exposed to TA6V, CSZ, YSZ, and Ca-SZ coating after 21 days under osteogenic conditions (Figure 12). The crystals appear as densely arranged blocks, with distinct morphological aspects and marked fractal details that are particularly visible at ×101 and ×500 magnification on the surface of each biomaterial. The crystals have a rough, irregular surface and rounded edges. Branched formations underlying the blocks of crystals, similar to roots or fractals, are clearly visible on all the biomaterials at various magnifications. These root-like features, indicated by the yellow arrows, appear to emerge beneath the main crystal blocks, suggesting structures that are expanding as they grow (Figure 12f,i–l). The fractal formations observed could reflect a hierarchical organization of HA crystals at multiple sequential levels, mediated by complex crystal growth dynamics. Relatively speaking, the abundance of HA crystals is very marked on the surface of Ca-SZ, CSZ, and YSZ, in contrast to the surface of TA6V.

### 3.8. EDS Mapping

Energy dispersive spectroscopy (EDS) mappings of HA crystals obtained after Saos-2 cell culture exposed to TA6V (Figure 13), CSZ (Figure 14), YSZ (Figure 15), and the Ca-SZ coating (Figure 16) reveal a heterogeneous distribution of chemical elements within the crystalline structures. The main elements making up the HA molecule of formula Ca_5_(PO_4_)_3_(OH) such as calcium (Ca), phosphorus (P), and oxygen (O) have been identified and confirmed, with the exception of hydrogen due to its very low atomic number. However, the presence of traces of sodium (Na) and chlorine (Cl) in the exact position of the branched formations described above and of potassium (K) diffusely throughout the biomaterials should be emphasized.

## 4. Discussion

The development of calcium-doped zirconia (Ca-SZ) coatings is an innovative approach aimed at overcoming the intrinsic limitations of current implant biomaterials. Indeed, Ca-SZ offers significant advantages compared with TA6V and other stabilized zirconia, both in terms of biocompatibility and bioactivity, in the light of the observations made in this study. Cell viability and proliferation kinetics on two Saos-2 and HaCat cell lines, as well as the molecular analyses carried out, clearly showed marked effects of Ca-SZ, favorably influencing cell viability and proliferation, as well as modulating the synthesis of pro-inflammatory cytokines by exposed cells. These synergistic properties simultaneously promote an environment conducive to bone regeneration and limit the onset of inflammatory responses, two essential factors for long-term osseointegration.

Ca-SZ rapidly induced viability and proliferation of Saos-2 and HaCat cells compared to the reference materials, as indicated by kinetic assays using WST-1 and CyQUANT NF. It is important to emphasize that in vitro cell proliferation and viability are essential and indispensable indicators of the biomaterial’s ability to integrate and promote the synthesis of newformed bone matrix. In this sense, calcium, due to its nature and function as a bioactive ion in various metabolic pathways, plays a dynamic role by influencing cell signaling and facilitating the formation of a bioactive microenvironment around the implant by chemotaxis directed via the activation of osteocytes [20].

In addition, scanning electron microscopy analyses of cell morphology revealed optimal cell adhesion on Ca-SZ, showing cell surfaces covered with multiple spicules and microvilli and the presence of significant cytoplasmic extensions such as pseudopodia and filopodia, reflecting close interaction with the surface. This suggests that calcium could influence the expression of adhesion molecules specific for strong cell adhesion and the establishment of strong intercellular connections, thereby reinforcing cell attachment and the bone–implant interface by dynamic involvement in the BMP, Wnt, and Ca^2+^/PKC activation pathways [21,22]. These cell projections are associated with adhesion proteins such as integrin, which facilitate adhesion and cell signaling, among other things [23]. However, observations of the cytoskeletal actin by confocal microscopy confirmed the morphological observations made in SEM micrographs, in particular the presence of a dense network of actin filaments oriented longitudinally and parallel to the surface of the Ca-SZ coating, similar to the reference biomaterials. This observation reflects the integrity of the cytoskeleton, which is essential for maintaining cell adhesion and propagation. The filamentous organization of actin observed is typical of well-anchored cells, showing directional polarization indicative of active spreading and particularly dynamic cell migration [24]. This abundantly developed actin network is crucial in the osteoblastic differentiation process, stabilizing cell morphology while actively facilitating the mechanosensitive signaling associated with osseointegration in contact with biomaterials [25,26,27].

In addition, analysis of the synthesis of pro-inflammatory cytokines by measuring TNF-α and IL-6 over a long-term period of 21 days clearly shows that the Ca-SZ coating induces a significant minimum reduction in these pro-inflammatory cytokines compared with certain reference biomaterials, in this case TA6V, despite already having levels of synthesis well below the level required to trigger an acute inflammatory response. This suggests a good level of tolerance and low immune activation. In this sense, it has been documented that calcium plays an immunomodulatory role, contributing to the regulation of the NF-κB pathway, thereby reducing the levels of synthesis of pro-inflammatory cytokines such as TNF-α and IL-6 and, consequently, reducing inflammation. In addition, the observation of low induction and reduction of TNF-α synthesis over time could indicate modulation of NF-κB pathway activation, suggesting that calcium from Ca-SZ contributes to an anti-inflammatory environment, as demonstrated by the work of Choi et al. [28]. From a purely fundamental point of view, such modulation is essential to minimizing the risk of osseointegration failure due to cellular damage caused by ROS, including cellular DNA strand breaks in an inflammatory microenvironment, given the close link between the NF-κB pathway and TNF-α [29,30]. In the light of this, it should be noted that these observations make the Ca-SZ coating a particularly attractive alternative biomaterial compared with reference materials with a view of minimizing the risk of inflammatory complications following the implant placement.

However, analyses in an osteogenic environment have shown that Ca-SZ offers early pro-osteogenic conditions that are particularly favorable to the synthesis of soluble collagens in a significantly dominant manner, while remaining relatively stable over time compared with reference biomaterials, which have lower and highly fluctuating levels of synthesis. Collagen is the main organic constituent of the extracellular bone matrix that supports all bone tissue. Consequently, good collagen synthesis is a key factor in osseointegration [31]. However, these results corroborate the work of Li et al., who argue that calcium ions promote osteogenic differentiation and collagen synthesis via the TGF-β1 pathway [32]. In parallel, long-term osteogenic assays have also shown that Ca-SZ promotes the rapid production of osteogenic markers such as alkaline phosphatase, osteocalcin, and osteopontin at high levels. It should be noted that ALP is an early marker of osteoblastic differentiation, reflecting the cells’ commitment to bone matrix synthesis, hence its observed progressive decrease over time, whereas osteocalcin and osteopontin are mainly late markers associated with bone maturation and mineralization, and the whole may be inter-regulated [33,34]. Ca-SZ significantly enhances the production of these molecules compared with other reference materials, including controls on D7 for osteopontin and on D21 for osteocalcin. However, there was a progressive increase in osteopontin over time for all the biomaterials and the opposite trend for osteocalcin, suggesting that the latter could be inter-regulated as previously reported in the literature [34]. To better understand the molecular interactions involved in cellular sensitivity to calcium ions, it is important to note that bone cells possess extracellular calcium-sensitive receptors (CaSRs). These receptors may play an extremely important role in maintaining extracellular calcium homeostasis and as essential mediators of the process of bone remodeling by directed chemotaxis [35,36].

In parallel with the quantification of osteogenic differentiation markers, the formation of HA crystals after 21 days of cell culture under osteogenic conditions was observed using SEM micrographs on Ca-SZ, including reference biomaterials. These crystals are rough and polyform in appearance, with rounded ends. Branched formations underlying the fractal-like blocks of crystals are clearly visible at high magnification. Relative as it may be, the abundance of HA crystals seems very marked on the surface of Ca-SZ, CSZ, and YSZ, in contrast to the surface of TA6V. The fractal formations observed could reflect a hierarchical organization of HA crystals at multiple sequential levels, mediated by complex nucleation and crystal growth dynamics [37].

The presence of chemical elements making up the HA molecule with the formula Ca_5_(PO_4_)_3_(OH) was confirmed by EDS mapping analysis on crystals. The presence of Na, Cl, and K within HA crystals is notable and could be attributed to inclusions of mineral salts present in the culture medium or to ionic diffusion processes during crystal growth under osteogenic conditions. Branched HA formations could indicate preferential growth pathways and selective ion incorporation mechanisms. HA is highly reactive and is able to tolerate certain ionic substitutions, such as K, Na, and Cl, present in the formation environment due to its relatively flexible crystalline structure, which can give rise to analogous minerals such as chlorapatite [38,39]. HA plays an extremely important role in the osseointegration process, as reported by Chamrad et al. in the context of cranial implantation [40]. Other authors report the favorable effects of HA on peri-implant bone neoformation, leading to rapid osseointegration [41]. It should be noted that all the biomaterials tested, including the Ca-SZ coating, present a set of optimal pro-osteogenic conditions that are particularly favorable to bone mineralization through cellular maturation processes, leading to the production of the mineralized extracellular matrix. Ultimately, by progressively reducing the synthesis of pro-inflammatory cytokines while actively supporting the regulation of osteogenic markers in the long term, the unique effect of calcium from Ca-SZ goes beyond simply playing a role in cell signaling and cytokine modulation. The fundamental aspect is its key role in oriented chemotaxis, which is the primordial trigger. The consequence would be optimal bone regeneration, giving Ca-SZ exceptional potential for the development of new biomaterials for endosseous implants.

However, while the overall differences observed between the calcium-free TA6V and the Ca-SZ coating in favor of the latter are clear and consistent, highlighting the better biological properties of the Ca-SZ thanks to the addition of calcium, the parallel is less obvious compared with the CSZ bulk material, which also contains calcium. In fact, intrinsic mechanisms linked to their difference in structure and their difference in elemental composition could explain the biological differences observed between the bulk ZrO/CaO-95/5w% (CSZ) material and the Ca-SZ coating composed of approximately 31% zirconium, 5% calcium, and 63% oxygen, predisposing the latter to a potentially higher reactivity of the surface oxides, which could contribute to improving bioactivity. In addition, the polycrystalline microstructure of the Ca-SZ coating, which can be characterized by an increased number of grain boundaries due to an inter-relation between this characteristic and the PVD synthesis method, could promote dynamic ionic interactions at the cell–material interface, making calcium ions more accessible to cells and enabling them to interact with biological signaling pathways, which are essential for osteogenic differentiation and better modulation of pro-inflammatory cytokines, such as the significant reduction in TNF-alpha over the long term.

Nevertheless, it has been reported in the literature that titanium and zirconia particles can induce cytotoxicity in HGF and THP-1 cells [42]. This information raises questions about the nature and form of biomaterials in contact with living organisms, particularly particulate forms whose cytotoxicity is often reported. Nevertheless, caution is called for regarding the clinical application of Ca-SZ. Comprehensive, in-depth transcriptomic studies could provide ample information about the influence of Ca-SZ on the differential expression of genes encoding proteins involved in various metabolic pathways that are important both in the short and long term for drawing up a complete biological profile of this material. On another level, Raman spectroscopy will make it possible to identify the chemical composition of the surface oxide layer of the Ca-SZ coating in order to provide clues as to the complex ionic interactions between the Ca-SZ and the cells. Finally, it would be rewarding to quantify the potential release of these ions by ICP-MS in an aqueous medium.

## 5. Conclusions

The aim of this study was to assess the biocompatibility of the Ca-SZ coating compared with reference biomaterials in order to identify its contribution to the synthesis of osteogenic markers and the inflammatory response in vitro, two issues that are particularly fundamental to optimal osseointegration. The results show that Ca-SZ promotes cell viability and proliferation, as well as the synthesis of osteogenic markers, while inducing a moderate inflammatory response. The uniqueness of this finding lies in the dynamic involvement of calcium from this biomaterial in cell signaling mechanisms, which may promote osseointegration while reducing inflammatory complications. Here, the Ca-SZ coating appears to be a serious alternative with superior overall biological properties to reference biomaterials. This work, therefore, proposes a new approach, positioning Ca-SZ as a promising candidate for hybrid implants.

## Figures and Tables

**Figure 1 jfb-16-00191-f001:**
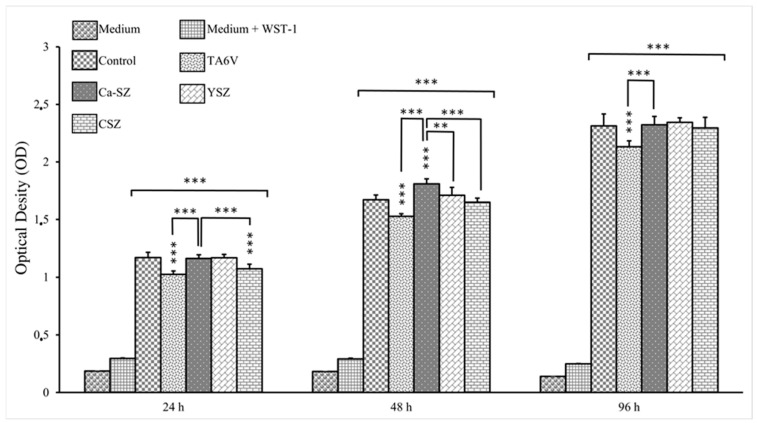
Kinetics of cell viability by WST-1 of Saos-2 cells exposed to CSZ, TA6V, and YSZ biomaterials and Ca-SZ coating compared to different exposure times of 24 h, 48 h, and 96 h. ** for 0.01 ≤ *p* < 0.05, and finally *** for 0.001 and below ≤ *p* < 0.01.

**Figure 2 jfb-16-00191-f002:**
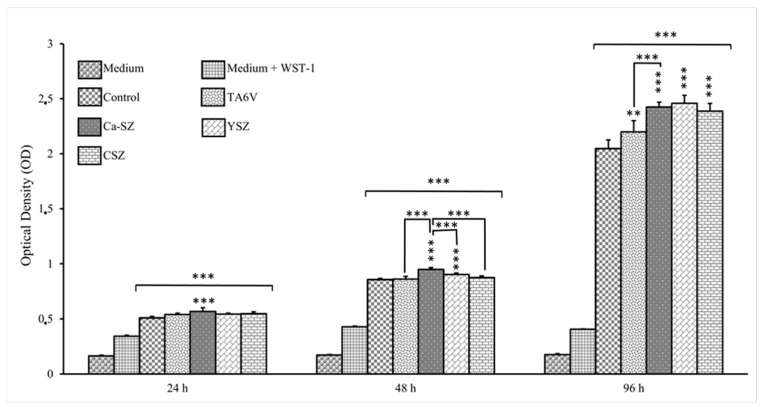
Kinetics of viability by WST-1 of HaCaT 2 cells exposed to CSZ, TA6V, and YSZ biomaterials and Ca-SZ coating compared to different exposure times of 24 h, 48 h, and 96 h. ** for 0.01 ≤ *p* < 0.05, and finally *** for 0.001 and below ≤ *p* < 0.01.

**Figure 3 jfb-16-00191-f003:**
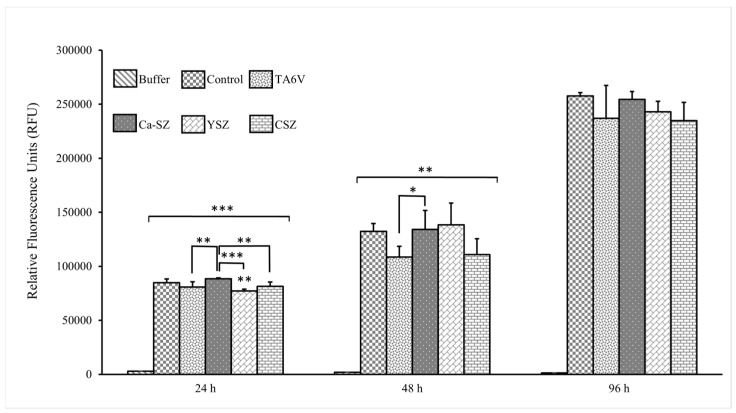
CyQUANT NF proliferation kinetics of Saos-2 cells exposed to CSZ, TA6V, and YSZ biomaterials and Ca-SZ coating compared to different exposure times of 24 h, 48 h, and 96 h. * for 0.05 ≤ *p* < 0.1; ** for 0.01 ≤ *p* < 0.05, and finally *** for 0.001 and below ≤ *p* < 0.01.

**Figure 4 jfb-16-00191-f004:**
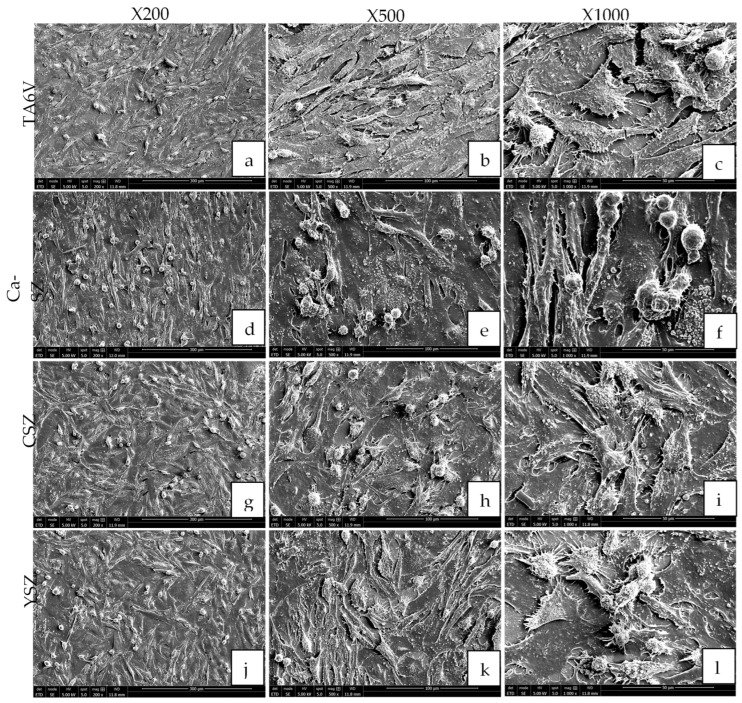
The morphology of Saos-2 cells on TA6V (**a**–**c**), CSZ (**g**–**i**), YSZ (**j**–**l**), and Ca-SZ coating (**d**–**f**) surfaces at ×200, ×500, and ×1000 magnification after the 96 h exposure.

**Figure 5 jfb-16-00191-f005:**
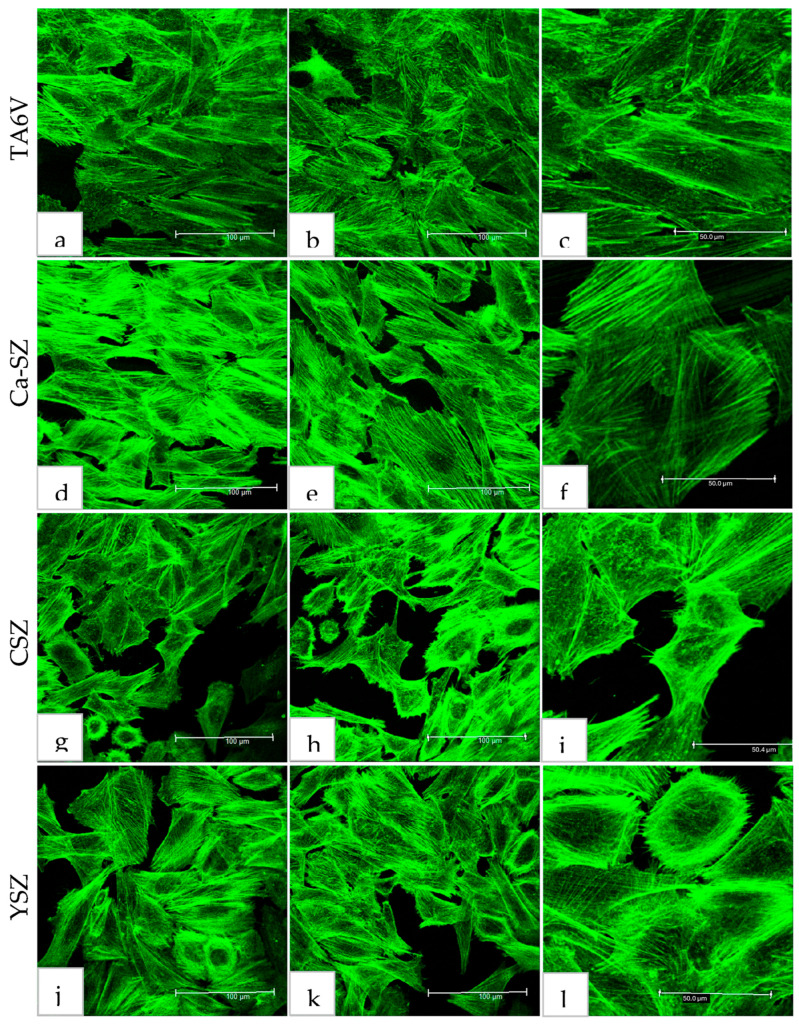
Actin cytoskeleton organization of Saos-2 cells after 72 h culture on TA6V (**a**–**c**), CSZ (**g**–**i**), YSZ (**j**–**l**), and Ca-SZ coating (**d**–**f**).

**Figure 6 jfb-16-00191-f006:**
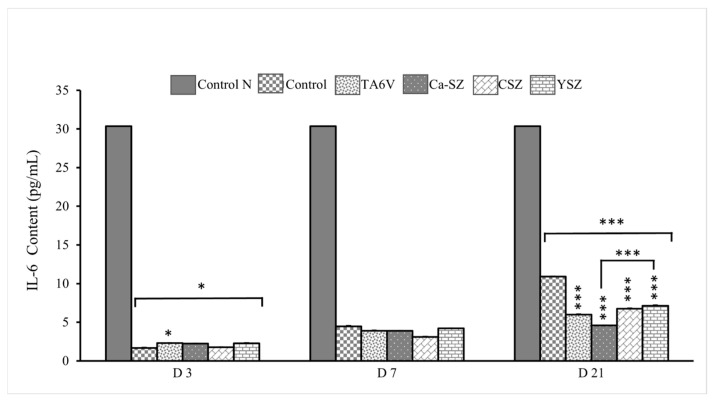
IL-6 evolution of IL-6 concentrations from supernatants of Saso-2 cells exposed to TA6V, CSZ, YSZ, and Ca-SZ coating on D3, D7, and D2. Control N is a reference controlecommended by the manufacturer of the KIT LISA Invitrogen. * for 0.05 ≤ *p* < 0.1; *** for 0.001 and below ≤ *p* < 0.01.

**Figure 7 jfb-16-00191-f007:**
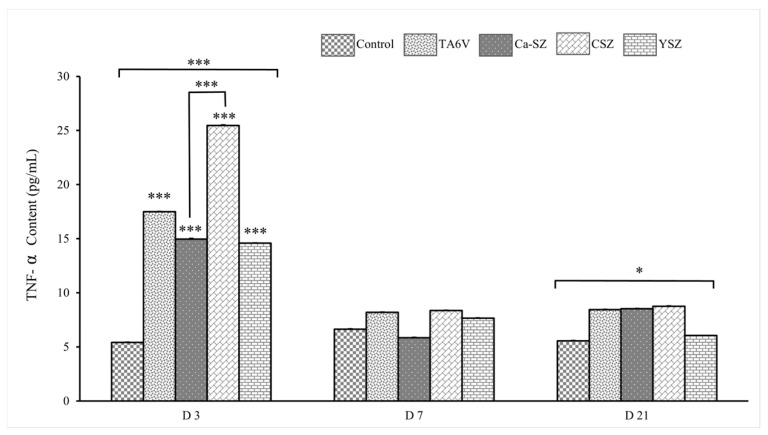
Evolution of TNF-α concentrations from supernatant of Saso-2 cells exposed to TA6V, CSZ, YSZ, and Ca-SZ coating on D3, D7, and D2. * for 0.05 ≤ *p* < 0.1; *** for 0.001 and below ≤ *p* < 0.01.

**Figure 8 jfb-16-00191-f008:**
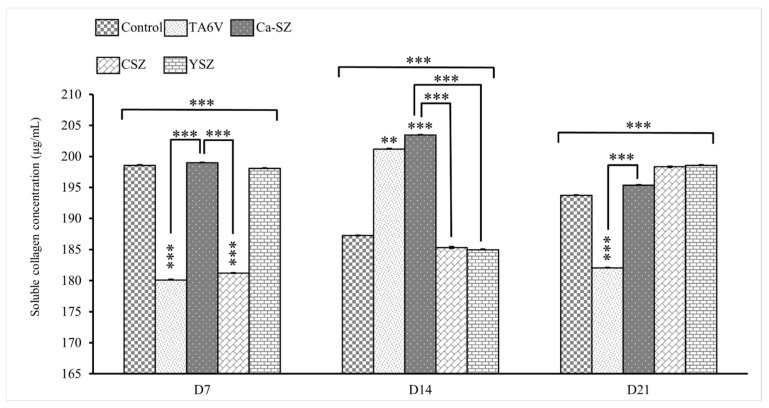
Evolution of soluble collagen concentrations in cell supernatant from TA6V-exposed Saos-2, CSZ, YSZ, and Ca-SZ coatings on days 7, 14, and 21. ** for 0.01 ≤ *p* < 0.05, and finally *** for 0.001 and below ≤ *p* < 0.01.

**Figure 9 jfb-16-00191-f009:**
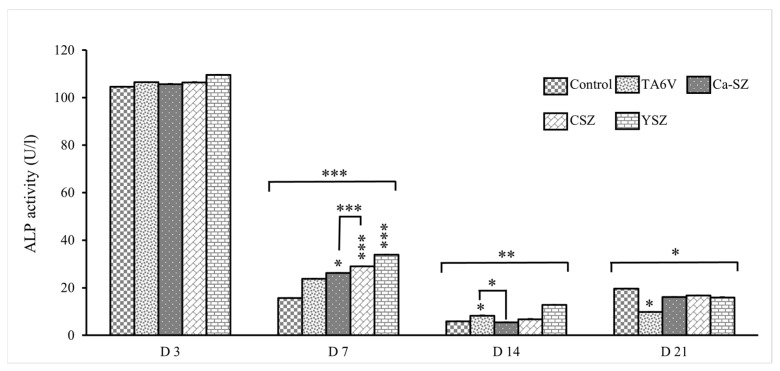
Evolution of alkaline phosphatase activity in Saos-2 cells exposed to TA6V, CSZ, YSZ, and Ca-SZ coating on days 3, 7, 14, and 21. * for 0.05 ≤ *p* < 0.1; ** for 0.01 ≤ *p* < 0.05, and finally *** for 0.001 and below ≤ *p* < 0.01.

**Figure 10 jfb-16-00191-f010:**
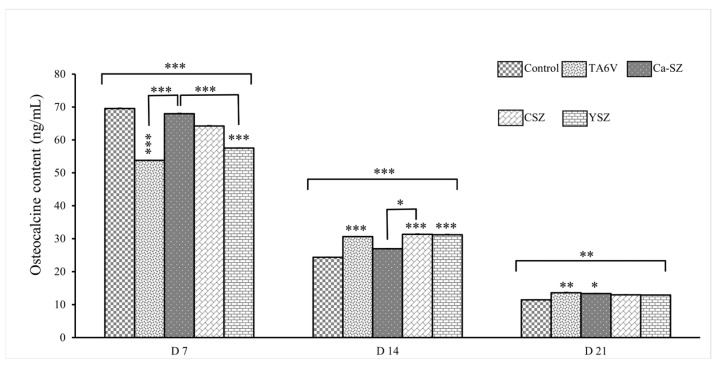
Evolution of osteocalcin concentration in the cell supernatant of Saos-2 cells exposed to TA6V, CSZ, YSZ, and the Ca-SZ coating on days 7, 14, and 21. * for 0.05 ≤ *p* < 0.1; ** for 0.01 ≤ *p* < 0.05, and finally *** for 0.001 and below ≤ *p* < 0.01.

**Figure 11 jfb-16-00191-f011:**
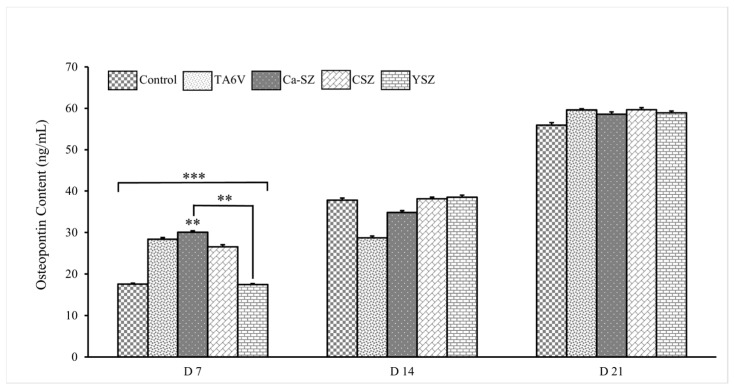
Osteopontin concentration in cell supernatants from Saos-2 exposed to TA6V, CSZ, YSZ, and Ca-SZ coating on days 7, 14, and 21. ** for 0.01 ≤ *p* < 0.05, and finally *** for 0.001 and below ≤ *p* < 0.01.

**Figure 12 jfb-16-00191-f012:**
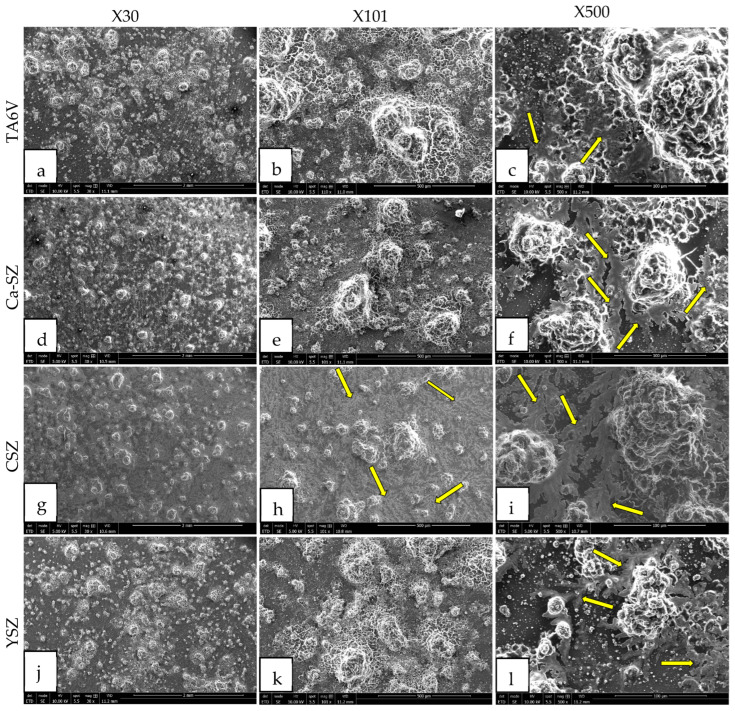
SEM micrographs of hydroxyapatite crystals formed on the surface of TA6V (**a**–**c**), CSZ (**g**–**i**), YSZ (**j**–**l**), and Ca-SZ (**d**–**f**) coating after differentiation of Saos-2 cells under osteogenic conditions for 21 days. Yellow arrows indicate fractal formations of hydroxyapatite crystals.

**Figure 13 jfb-16-00191-f013:**
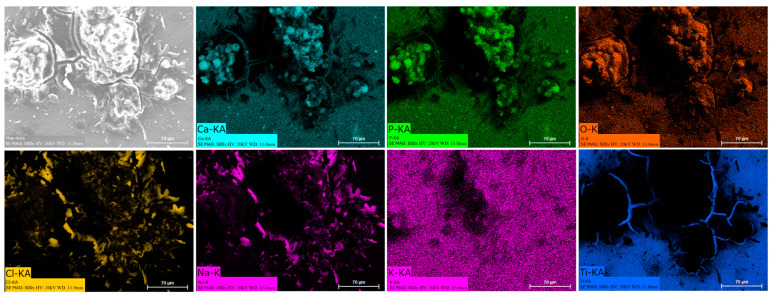
EDS mapping of hydroxyapatite crystals differentiated from Saos-2 cells on TA6V surface under osteogenic conditions for 21 days.

**Figure 14 jfb-16-00191-f014:**
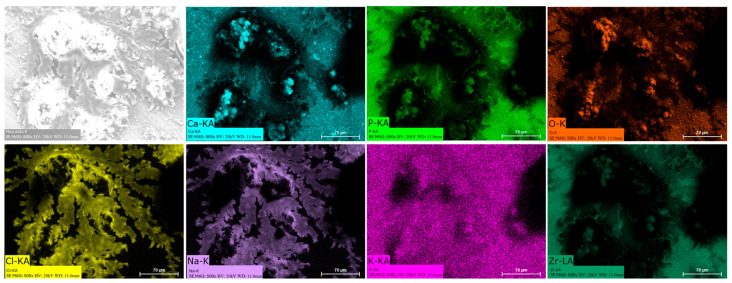
EDS mapping of hydroxyapatite crystals differentiated from Saos-2 cells on a Ca-SZ surface under osteogenic conditions for 21 days.

**Figure 15 jfb-16-00191-f015:**
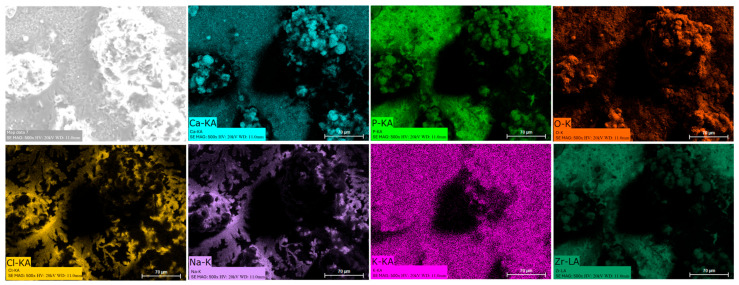
EDS mapping of hydroxyapatite crystals differentiated from Saos-2 cells on CSZ surface under osteogenic conditions for 21 days.

**Figure 16 jfb-16-00191-f016:**
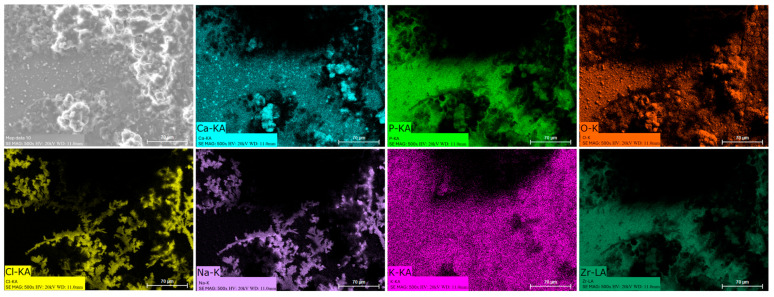
EDS mapping of hydroxyapatite crystals differentiated from Saos-2 cells on the YSZ surface under osteogenic conditions for 21 days.

## Data Availability

The original contributions presented in the study are included in the article, further inquiries can be directed to the corresponding author.

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
