# Peer review of "Towards Enhanced Osteointegration: A Comparative and In-Depth Study of the Biocompatibility of an Innovative Calcium-Doped Zirconia Coating for Biomedical Implants"

_jfb, 2025, doi:10.3390/jfb16060191_

Round 1

Reviewer 1 Report

Comments and Suggestions for Authors

The manuscript titled " Towards Enhanced Osteointegration: A Comparative and In-Depth Study of the Biocompatibility of an Innovative Calcium-Doped Zirconia Coating for Biomedical Implants " addresses a critical clinical challenge, peri-implantitis and osseointegration failure, by proposing a novel Ca-SZ coating. This aligns with current trends in biomaterial innovation for dental implants. However, several key areas require significant improvement before the manuscript can be considered for publication.

  1. While concise, the abstract lacks specific details on sample size (e.g., N=3 biological replicates) and statistical methods. Explicitly state sample sizes and statistical thresholds (e.g., “p<0.05”) to enhance clarity.
  2. Redundancy in keywords (“osseointegration” appears twice). Replace one instance with a broader term (e.g., “biomaterials”) to improve discoverability.
  3. The rationale for using PVD to deposit Ca-SZ is underdeveloped. Additionally, some references (e.g., peri-implantitis prevalence rates) lack contextualization. Expand on why PVD was chosen over other coating methods (e.g., adhesion strength, surface topography benefits). Update references for epidemiological data (e.g., recent meta-analyses on peri-implantitis prevalence).

4.While the literature review emphasizes experimental and biochemical aspects of osseointegration, it could further acknowledge the growing integration of computational modeling in implant biomaterial evaluation. Recent studies employing finite element analyses have demonstrated the utility of simulating bone remodeling dynamics around functionally graded implants, offering insights into stress distribution and mechanobiological interactions that complement in vitro findings. Such approaches could strengthen the clinical translatability of this work by predicting long-term biomechanical stability and guiding implant design optimizations. For instance, the work available could be incorporated to strengthen the introductory framework (https://doi.org/10.1002/cnm.3875 ; https://doi.org/10.1002/cnm.3750)

  1. The study’s experimental design is robust, employing a comprehensive array of assays (cell viability, proliferation, cytokine profiling, osteogenic markers) and advanced imaging (SEM, confocal microscopy, EDS mapping) to evaluate biocompatibility. The use of both Saos-2 (osteoblastic) and HaCaT (epithelial) cells strengthens relevance by assessing responses across tissue types. However, critical details are lacking, including cell line passage numbers and validation of ethical compliance for cell use, which limits reproducibility. Statistical assumptions (e.g., normality testing for ANOVA) are not explicitly addressed, and the rationale for control groups (such as the absence of a sham-coated or uncoated surface comparator) is unclear, potentially undermining interpretation of the Ca-SZ coating’s specific effects. To improve rigor, the authors should specify passage numbers, confirm adherence to ethical guidelines, detail statistical assumptions, and justify control group selection (e.g., positioning TA6V as a baseline reference). These clarifications would enhance methodological transparency and strengthen the validity of comparative analyses.
  2. SEM images lack scale bars, and EDS mapping is qualitative without quantitative elemental analysis. Hydroxyapatite crystal quantification relies solely on morphological description. Include scale bars in SEM images and provide EDS quantitative data (e.g., weight percentages of Ca, P). Use X-ray diffraction (XRD) to confirm HA crystallinity.
  3. The comparison between Ca-SZ and CSZ (bulk material) is superficial, despite their structural/compositional differences. Cytotoxicity of released ions (noted in prior literature) is not addressed. Discuss how Ca-SZ’s polycrystalline structure and surface oxide reactivity differ from CSZ. Address potential ion release risks (e.g., Zr or Ca ions) and cite relevant studies (e.g., Schwarz et al., 2019, on titanium/zirconia particle toxicity).
  4. Conclusion overstates clinical applicability without acknowledging in vitro limitations (e.g., absence of immune cells, mechanical loading). Emphasize the need for in vivo validation and long-term toxicity studies before clinical translation.

The manuscript is recommended for publishing provided the above comments are adequately addressed.

Author Response

We sincerely thank the reviewers for their insightful comments and constructive suggestions, which have greatly contributed to improving the quality and clarity of our work.Their expertise was invaluable in refining our manuscript.

  1. While concise, the abstract lacks specific details on sample size (e.g., N=3 biological replicates) and statistical methods. Explicitly state sample sizes and statistical thresholds (e.g., “p<0.05”) to enhance clarity.

Response 1 : Done

  1. Redundancy in keywords (“osseointegration” appears twice). Replace one instance with a broader term (e.g., “biomaterials”) to improve discoverability.

Response 2 : This is not a redundancy; the terms “osteointegration” and “osteogenesis” refer to two distinct biological concepts.

  1. The rationale for using PVD to deposit Ca-SZ is underdeveloped. Additionally, some references (e.g., peri-implantitis prevalence rates) lack contextualization. Expand on why PVD was chosen over other coating methods (e.g., adhesion strength, surface topography benefits). Update references for epidemiological data (e.g., recent meta-analyses on peri-implantitis prevalence).

Response 3 : The central focus of this manuscript lies in the biological aspects and the influence of the innovative Ca-SZ coating on cellular parameters assessed in vitro. While the choice of deposition technique is indeed relevant, it is not the primary objective of this study. However, this aspect has already been thoroughly addressed in a previous publication dedicated to the microstructural and mechanical characterization of the coating. That study included detailed analyses using SEM, TEM, and SEM-FIB, as well as mechanical evaluations (nanoindentation and scratch testing) to validate the coating’s mechanical performance and support the selection of the PVD technique. This choice was justified by the excellent adhesion of thin films, high reproducibility, compatibility with metallic substrates, and the ability to finely control the microstructure.

Regarding the epidemiological data on peri-implantitis, the references cited in the manuscript are up to date and aligned with recent literature. The trends reported are consistent with recent meta-analyses, which confirm a variable but significant prevalence of peri-implantitis depending on diagnostic criteria.

  1. While the literature review emphasizes experimental and biochemical aspects of osseointegration, it could further acknowledge the growing integration of computational modeling in implant biomaterial evaluation. Recent studies employing finite element analyses have demonstrated the utility of simulating bone remodeling dynamics around functionally graded implants, offering insights into stress distribution and mechanobiological interactions that complement in vitro findings. Such approaches could strengthen the clinical translatability of this work by predicting long-term biomechanical stability and guiding implant design optimizations. For instance, the work available could be incorporated to strengthen the introductory framework (https://doi.org/10.1002/cnm.3875 ; https://doi.org/10.1002/cnm.3750)

Response 4 : We consider that the finite element modeling approach applied to implant evaluation is highly relevant and complementary to in vitro analyses. Although this is not included in the current manuscript, we plan to explore this direction in ongoing studies under development, with the aim of broadening the scope of our analyses and enhancing the clinical translatability of our findings

  1. The study’s experimental design is robust, employing a comprehensive array of assays (cell viability, proliferation, cytokine profiling, osteogenic markers) and advanced imaging (SEM, confocal microscopy, EDS mapping) to evaluate biocompatibility. The use of both Saos-2 (osteoblastic) and HaCaT (epithelial) cells strengthens relevance by assessing responses across tissue types. However, critical details are lacking, including cell line passage numbers and validation of ethical compliance for cell use, which limits reproducibility. Statistical assumptions (e.g., normality testing for ANOVA) are not explicitly addressed, and the rationale for control groups (such as the absence of a sham-coated or uncoated surface comparator) is unclear, potentially undermining interpretation of the Ca-SZ coating’s specific effects. To improve rigor, the authors should specify passage numbers, confirm adherence to ethical guidelines, detail statistical assumptions, and justify control group selection (e.g., positioning TA6V as a baseline reference). These clarifications would enhance methodological transparency and strengthen the validity of comparative analyses.

Response 5 : We thank you for this relevant comment. Given the design of the study, we initially did not consider it necessary to specify the cell passage numbers. However, we have now included this information in the manuscript for greater clarity.

Regarding ethical considerations, we confirm that all procedures involving the use of cell lines were carried out in strict compliance with the applicable regulations, as previously stated. Therefore, there is no concern regarding ethical compliance or the reproducibility of the study.

Finally, the justification for the control groups and reference biomaterials has been explicitly detailed in the “Sample Preparation and Sterilisation” section, where we thoroughly explain the rationale behind the choice of each reference material.

  1. SEM images lack scale bars, and EDS mapping is qualitative without quantitative elemental analysis. Hydroxyapatite crystal quantification relies solely on morphological description. Include scale bars in SEM images and provide EDS quantitative data (e.g., weight percentages of Ca, P). Use X-ray diffraction (XRD) to confirm HA crystallinity.

Response 6 : We thank you for this remark. We would like to clarify that scale bars are indeed present in all SEM images, clearly visible at the bottom right within the black frame of each image.

Regarding the EDS mapping, our aim was not to conduct a quantitative elemental analysis, but rather to confirm the presence of hydroxyapatite crystals through elemental identification (presence of Ca and P). Indeed, accurate elemental quantification by EDS has significant limitations due to its limited spatial resolution and low sensitivity for superficial or heterogeneous layers.

For a more rigorous quantification of mineralization or hydroxyapatite deposits, more appropriate approaches should have been used, such as Alizarin Red S staining, combined with digital image analysis to quantify the surface area or intensity of the deposits. These methods will be incorporated into our ongoing complementary studies.

  1. The comparison between Ca-SZ and CSZ (bulk material) is superficial, despite their structural/compositional differences. Cytotoxicity of released ions (noted in prior literature) is not addressed. Discuss how Ca-SZ’s polycrystalline structure and surface oxide reactivity differ from CSZ. Address potential ion release risks (e.g., Zr or Ca ions) and cite relevant studies (e.g., Schwarz et al., 2019, on titanium/zirconia particle toxicity).

Response 7 : We thank you for this valuable comment. We would like to clarify that we did not intend to draw definitive conclusions regarding the structural and compositional differences between Ca-SZ (coating) and CSZ (bulk material), as no in-depth characterization was specifically performed on the bulk CSZ material. A rigorous comparison would require additional analyses, including X-ray diffraction (XRD), scanning electron microscopy (SEM), and X-ray photoelectron spectroscopy (XPS), carried out directly on the bulk CSZ samples. For this reason, we have addressed this topic only as a plausible hypothesis, suggesting it as a potential line of reflection rather than a formally demonstrated difference. These assumptions remain to be verified

  1. Conclusion overstates clinical applicability without acknowledging in vitro limitations (e.g., absence of immune cells, mechanical loading). Emphasize the need for in vivo validation and long-term toxicity studies before clinical translation.

Response 8 :  We thank you for this relevant comment. We would like to emphasize that the limitations of our in vitro study have been thoroughly discussed in the “Discussion” section. In the same section, we clearly stated the need for further analyses, particularly transcriptomic studies and in vivo investigations, to better assess tissue responses, long-term toxicity, and the clinical translatability of our Ca-SZ coating. We therefore fully share your perspective and have taken care to incorporate these considerations into the discussion.

Reviewer 2 Report

Comments and Suggestions for Authors

This paper investigates the biocompatibility of calcium-doped Zirconia coating on titanium alloy. The biocompatibility includes cellular viability and proliferation, synthesis of pro-inflammatory cytokine, and osteogenic differentiation. Following points should be addressed before paper re-submission.

Major

1) In this study, cells were seeded on the substrates (TA6V, CSZ, YSZ, Ca-SZ) placed on the 12 well plate. To avoid the influence of cells attached on the plate, polystyrene, substrates were usually transferred to fresh 12 well plate after initial attachment in most literatures. Did the authors do that?

2) Figure 6: I think it difficult for readers to understand the meaning and necessity of “Control N” group. More detailed explanation should be added.

3) In this study, the biocompatibility was demonstrated from various viewpoints. However, little experiments to elucidate mechanisms were performed. It would be better to add data on mechanism although I understand the authors discussed the mechanism by quoting literatures.

Minor

1) Line 25: “PAL” should be “ALP”.

2) Line 383: “3.6.1.” would be better to be added.

  Line 401: “3.7.” would be better to be changed to “3.6.2.”.

  Line 419: “3.7.1.” would be better to be changed to “3.6.3”.

  Line 435: “3.7.2.” would be better to be changed to “3.6.4”.

  Line 450: “3.7.3.” would be better to be changed to “3.6.5”.

  Line 469: “3.8.” would be better to be changed to “3.6.6”.

Author Response

We sincerely thank the reviewers for their insightful comments and constructive suggestions, which have greatly contributed to improving the quality and clarity of our work.Their expertise was invaluable in refining our manuscript.

1) In this study, cells were seeded on the substrates (TA6V, CSZ, YSZ, Ca-SZ) placed on the 12 well plate. To avoid the influence of cells attached on the plate, polystyrene, substrates were usually transferred to fresh 12 well plate after initial attachment in most literatures. Did the authors do that?

 Response 1 : We thank the reviewer for this insightful comment. In our study, the transfer step was not performed because the discs were specifically dimensioned to fully cover the bottom surface of the 12-well plates. This deliberate choice ensured that cells could only adhere to the biomaterial surfaces, thereby preventing interference from polystyrene adhesion. By carefully matching the disc size to the well dimensions, we aimed to simplify the experimental protocol while ensuring the reliability of surface-specific cellular responses.

2) Figure 6: I think it difficult for readers to understand the meaning and necessity of “Control N” group. More detailed explanation should be added.

 Response 2 : Thank you for your comment. A detailed explanation regarding the "Control N" group has been provided in the "Materials and Methods" section, specifically in the subsection "Determination of Pro-Inflammatory Cytokine Synthesis" (line 2017). We have outlined the rationale for including this group in our study. We hope this clarification will help provide a better understanding of its necessity

3) In this study, the biocompatibility was demonstrated from various viewpoints. However, little experiments to elucidate mechanisms were performed. It would be better to add data on mechanism although I understand the authors discussed the mechanism by quoting literatures.

 Response 3 : Thank you for this valuable remark. This study primarily relies on a factual demonstration of biocompatibility, based on the techniques mentioned (cell viability, proliferation, cytokine profiling, osteogenic markers, etc.) which allow for a direct and measurable evaluation of the coating's effects. However, we acknowledge that much deeper analyses, such as transcriptomic studies or other molecular approaches, as mentioned in the "Discussion" section, would be necessary to explore the underlying mechanisms in more detail and provide a comprehensive understanding of these observations. These analyses will be part of our future research.

Minor

  • Line 25: “PAL” should be “ALP”.

Response : Done

  • Line 383: “3.6.1.” would be better to be added.

Response : This is a reformatting of the journal; we will revisit it during the final revision.

  Line 401: “3.7.” would be better to be changed to “3.6.2.”.

Response : This is a reformatting of the journal; we will revisit it during the final revision.

  Line 419: “3.7.1.” would be better to be changed to “3.6.3”.

Response : This is a reformatting of the journal; we will revisit it during the final revision.

  Line 435: “3.7.2.” would be better to be changed to “3.6.4”.

Response : This is a reformatting of the journal; we will revisit it during the final revision.

  Line 450: “3.7.3.” would be better to be changed to “3.6.5”.

Response : This is a reformatting of the journal; we will revisit it during the final revision.

  Line 469: “3.8.” would be better to be changed to “3.6.6”.

Response : This is a reformatting of the journal; we will revisit it during the final revision

Round 2

Reviewer 1 Report

Comments and Suggestions for Authors

Accept in present form

Reviewer 2 Report

Comments and Suggestions for Authors

Since the responses are all fine, the reviewer agrees the publication of this manuscript in  Journal of Functional Biomaterials.